# Select before Act: Spatially Decoupled Action Repetition for Continuous Control

**Buqing Nie[1], Yangqing Fu[1], Yue Gao[1,2]***
[1]MoE Key Lab of Artificial Intelligence, AI Institute, Shanghai Jiao Tong University
[2]Shanghai Innovation Institute, Shanghai, P.R. China
{niebuqing,frank79110,yuegao}@sjtu.edu.cn

## Abstract

Reinforcement Learning (RL) has achieved remarkable success in various continuous control tasks, such as robot manipulation and locomotion. Different to mainstream RL which makes decisions at individual steps, recent studies have incorporated action repetition into RL, achieving enhanced action persistence with improved sample efficiency and superior performance. However, existing methods treat all action dimensions as a whole during repetition, ignoring variations among them. This constraint leads to inflexibility in decisions, which reduces policy agility with inferior effectiveness. In this work, we propose a novel repetition framework called **SDAR**, which implements **S**patially **D**ecoupled **A**ction **R**epetition through performing closed-loop *act-or-repeat* selection for each action dimension individually. SDAR achieves more flexible repetition strategies, leading to an improved balance between action *persistence* and *diversity*. Compared to existing repetition frameworks, SDAR is more sample-efficient with higher policy performance and reduced action fluctuation. Experiments are conducted on various continuous control scenarios, demonstrating the effectiveness of spatially decoupled repetition design proposed in this work.

## 1 Introduction

Recently, Deep Reinforcement Learning (DRL) (Sutton & Barto, 2018) has achieved remarkable success in continuous control domains, such as robot manipulation (Gu et al., 2017), locomotion (Lee et al., 2020b; Zhang et al., 2024), and autonomous driving (Kiran et al., 2021). Although conventional Reinforcement Learning (RL) algorithms have demonstrated significant potential across various applications, they typically make decisions at individual time steps, neglecting higher-level decision-making mechanisms and the temporal consistency of action sequences (Silver et al., 2014; Schulman et al., 2017; Haarnoja et al., 2018). This leads to inefficient exploration during training, challenging credit assignment tasks over long horizons, and poor sample efficiency (Dabney et al., 2021; Yu et al., 2021; Biedenkapp et al., 2021; Zhang et al., 2022; Lee et al., 2024).

One mainstream solution is Hierarchical RL (HRL) based on temporal abstraction (Sutton et al., 1999; Precup, 2000), where the high-level policy decomposes the task into simpler subgoals, and low-level policies are designed to solve isolated subtasks (Vezhnevets et al., 2017; Pateria et al., 2021). However, most HRL methods are task specific, which requires expert knowledge and handcraft design given different applications, such as pre-defining options in the option framework (Bacon et al., 2017; Nachum et al., 2018; Zhang et al., 2021). A simple yet effective strategy for low-level policies involves executing an action repeatedly for a number of steps, which has been actively explored and proven to be effective in various applications (Lakshminarayanan et al., 2017; Sharma et al., 2017; Dabney et al., 2021; Lee et al., 2024). Action repetition methods improve *action persistence* through *act-or-repeat* selections, making exploration trajectories more temporally consistent, leading to higher training efficiency and reduced fluctuations (Chen et al., 2021; Yu et al., 2021; Biedenkapp et al., 2021).

---

*Corresponding author.

However, as illustrated in Fig. 1, existing action-repetition approaches treat all action dimensions as a whole during *act-or-repeat* selection, disregarding differences among them. This constrains all action dimensions to repeat previous actions or make new decisions simultaneously at each step. This design reduces the effectiveness of action repetition, because different dimensions may be required to select repetition at different time steps, which is quite common in continuous control tasks (Kalyanakrishnan et al., 2021). For example, robotic systems typically consist of multiple controllers operating at different frequencies due to their system specification, corresponding to different repetition schema for each actuator (Raman et al., 2017; Lee et al., 2020a).

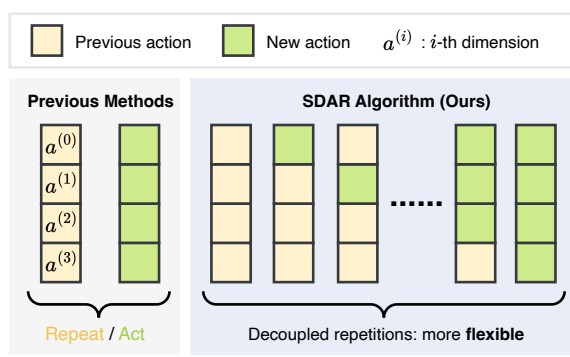

Figure 1: Difference between repetition strategies of previous methods (left) and SDAR (right). Our method achieves a more flexible strategy through the spatially decoupled repetition design.

To address this issue, we propose **S**patially **D**ecoupled **A**ction **R**epetition (SDAR), a new flexible action repetition framework for continuous control tasks. SDAR conducts closed-loop repetition for each action dimension individually, which is composed of two stages: *selection* and *action*. During *selection*, the agent selects whether the previous actions should be repeated for each dimension. Afterwards, the agent generates new decisions for dimensions where *act* is chosen in the previous stage. As shown in Fig. 1, different to previous methods, *act-or-repeat* selections in SDAR are performed in a decoupled manner, leading to more flexible repetition strategies. Compared to exsiting repetition frameworks, SDAR achieves an improved balance between action *persistence* and *diversity*, resulting in improved efficiency and superior performance with reduced action fluctuation.

The main contributions of this work are summarized as follows:

- We propose a novel action repetition framework called **SDAR** for continuous control tasks. SDAR implements **S**patially **D**ecoupled **A**ction **R**epetition through performing closed-loop *act-or-repeat* selection for each action dimension individually.

- Compared to previous repetition methods, SDAR offers more flexible repetition strategies, leading to improved balance between action *persistence* and *diversity*. This results in a higher sample efficiency with superior performance and reduced action fluctuation simultaneously.

- Experiments are conducted on various continuous control tasks, demonstrating the superior performance of our method on training efficiency and final performance. This demonstrates the effectiveness of the spatially decoupled framework for action repetition.

## 2 RELATED WORK

### 2.1 TEMPORAL ABSTRACTION

Temporal Abstraction is proposed in the semi-MDP formulation (Sutton et al., 1999; Precup, 2000) and commonly implemented based on the *options* framework (Stolle & Precup, 2002; Bacon et al., 2017; Harutyunyan et al., 2018). Each option describes a low-level policy and is defined as $\langle \mathcal{I}, \pi, \beta \rangle$, where $\mathcal{I}$ denotes the admissible states for the option initialization, $\pi$ is the policy that the option follows, and $\beta$ determines when the option is terminated. High-level policies are trained to solve tasks utilizing temporally extended actions provided in the options, rather than one-step actions without action persistence. Plenty of Hierarchical RL methods are proposed based on temporal abstraction, achieving faster exploration and higher sample efficiency in various sequential decision tasks (Lin et al., 2021; Yang et al., 2021). Some works are proposed to learn to design options through various techniques, including discovering state connectedness (Chaganty et al., 2012), replay buffer analysis (Eysenbach et al., 2019), and learning termination criteria (Vezhnevets et al., 2016; Harutyunyan et al., 2019). However, designing options is still a challenging task, which requires prior knowledge and handcraft tunning (Pateria et al., 2021; Yu et al., 2021; Lee et al., 2024).

## 2.2 ACTION REPETITION

One simple option strategy is repeating a primitive action for a number of steps, which is similar to the frame-skipping utilized in RL for video games (Bellemare et al., 2013; Braylan et al., 2015). Recently, action repetition has been actively researched and widely adopted in RL, which can achieve deeper exploration (Dabney et al., 2021), improve sample efficiency by reducing control granularity (Biedenkapp et al., 2021), and reduce action oscillations (Chen et al., 2021). Existing repetition works are classified as two categories: *open-loop* and *closed-loop* manners. Open-loop methods force the agent to repeat actions for a predicted number of steps without opportunity of early terminations, such as DAR (Lakshminarayanan et al., 2017), FiGAR (Sharma et al., 2017), TempoRL (Biedenkapp et al., 2021), and UTE (Lee et al., 2024). In contrast, closed-loop methods conduct *act-or-repeat* binary decision to decide if the previous action should be repeated, such as PIC (Chen et al., 2021) and TAAC (Yu et al., 2021). Compared to open-loop methods, closed-loop methods examine whether to repeat based on the current state, which is more flexible and improves performance in emergency situations. In this work, we propose a new closed-loop method to conduct act-or-repeat selections for each actuator individually given current state, which is more flexible and achieves higher action persistence with sufficient action diversity.

## 3 PRELIMINARIES

### 3.1 PROBLEM FORMULATION

In this work, we focus on model-free RL with continuous action space. The interaction process in RL is formulated as a Markov Decision Process (MDP), denoted as a tuple $\mathcal{M} = \langle \mathcal{S}, \mathcal{A}, P, R, \gamma \rangle$, where $\mathcal{S}$ is the state space, $\mathcal{A}$ is the action space, $P(s'|s, a)$ is the transition probability of the environment, $R : \mathcal{S} \times \mathcal{A} \to \mathbb{R}$ denotes the reward function, and $\gamma \in [0, 1)$ denotes the discount factor. The agent takes actions according to its policy, i.e. $a \sim \pi(\cdot|s)$. Our objective is to find a policy $\pi^*$ that maximizes the expected discounted return, i.e. $\pi^* = \arg\max_\pi \mathbb{E}_{a_t \sim \pi, s_{t+1} \sim P} \left[ \sum_{t=0}^{\infty} \gamma^t R(s_t, a_t) \right]$.

### 3.2 MODEL-FREE RL FOR CONTINUOUS CONTROL

In order to conduct policy evaluation, we define state-action value function $Q(s_t, a_t) = R(s_t, a_t) + \gamma \mathbb{E}_{\pi, P} \left[ \sum_{t'=t}^{\infty} \gamma^{t'} R(s_{t'}, a_{t'}) \right]$ as the discounted return starting from $s_t$, given that $a_t$ is taken and then $\pi$ is followed. The value function $V(s_t) = \mathbb{E}_{a_t \sim \pi} [Q(s_t, a_t)]$ denotes the discounted return starting from $s_t$ following $\pi$. Typically, both $Q$ and $V$ functions are modeled as neural networks, which are optimized using the Mean Square Error (MSE) loss, with target values obtained based on the Bellman equation. Take Soft Actor Critic (SAC) (Haarnoja et al., 2018) as an example, the policy is trained through the entropy augmented objective:

$$\pi' \leftarrow \arg\max_\pi \mathbb{E}_{s \sim \mathcal{D}, a \sim \pi} \left[ Q(s, a) - \alpha \log \pi(a|s) \right], \tag{1}$$

where $\mathcal{D}$ denotes the replay buffer, $\alpha$ is the temperature parameter, and $Q$ represents accumulated discounted reward augmented by the entropy.

## 4 METHODOLOGY

### 4.1 TWO-STAGE POLICY: SELECTION AND ACTION

As described in Fig. 2, the decision process of SDAR is composed of two stages. **(1) Selection:** choose which action dimensions to change previous decisions utilizing the selection policy $\beta$. **(2) Action:** generate new actions for action dimensions that choose *act* in the previous stage utilizing action policy $\pi$. Both two stages are described as follows.

#### 4.1.1 STAGE 1: SELECTION POLICY

As illustrated in the gray region of Fig. 2, given current state $s$ and the action $a^-$ at the previous step, we conduct *act-or-repeat* selection for each action dimension individually. Formally, $\beta(b|s, a^-) \in$

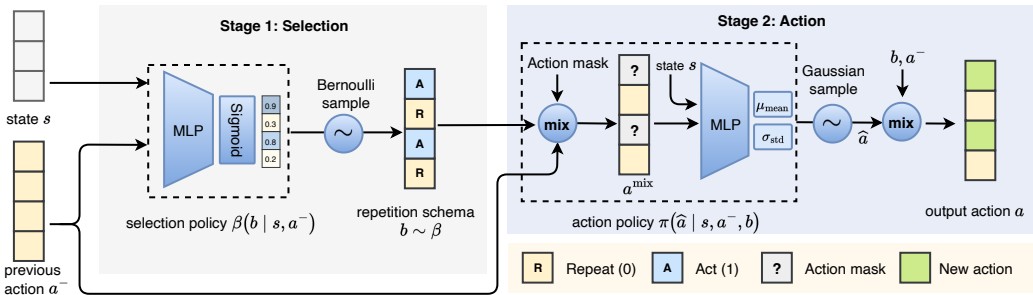

Figure 2: The two-stage decision process of SDAR algorithm. In the first stage (gray region), the selection policy $\beta$ makes *act-or-repeat* decision for each action dimension, determining whether the previous action $a^-$ (yellow blocks) should be repeated. In the second stage (blue region), the action policy $\pi$ generates new actions (green blocks) for dimensions that choose *act* in the first stage.

$[0,1]^{|A|}$, where $\beta_i$ denotes the probability of choosing *act* in the $i$-th action dimension. Afterwards, the repetition schema $b \in \{0,1\}^{|A|}$ at the current state is obtained by sampling on the Bernoulli distribution, i.e. $b_i \sim \beta_i(\cdot|s, a^-)$, where $b_i = 0$ and $b_i = 1$ indicate *repetition* and *action* of the $i$-th action dimension correspondingly.

### 4.1.2 STAGE 2: ACTION POLICY

As shown in the blue region of Fig. 2, the agent generates new actions for dimensions that choose *act* based on the current state $s$, the previous action $a^-$, and the repetition schema $b$. Firstly, we reduce redundant input dimensions through introducing the following Mix operation:

$$a^{\mathrm{mix}} = \mathrm{Mix}(b, a^-, \xi) = (1-b) \odot a^- + b \odot \xi, \tag{2}$$

where $\odot$ denotes element-wise hadamard product and $\xi$ denotes the action mask, which is a constant mask vector filled with meaningless values. In this work, $\xi_i = -2$ for $1 \le i \le |A|$ is utilized with $A = [-1, 1]^{|A|}$. This operation eliminates the information of previous actions $a^-$ at dimensions where *act* was chosen in the previous stage. This information is redundant and may disrupt the training process of $\pi$. Afterwards, we compute $\hat{a}$ utilizing action policy $\pi$:

$$\mu_{\mathrm{mean}}, \sigma_{\mathrm{std}} = \mathrm{MLP}(s, a^{\mathrm{mix}}), \quad \hat{a} = \mu_{\mathrm{mean}} + \sigma_{\mathrm{std}} \cdot n, \ \ n \sim \mathcal{N}(0, \mathcal{I}), \tag{3}$$

where $\mu_{\mathrm{mean}}$ and $\sigma_{\mathrm{std}}$ are the mean and standard derivation of the action distribution predicted by action policy $\pi$. The reparameterization trick is utilized to ensure that this process remains differentiable (Haarnoja et al., 2018). Finally, we obtain the output action $a$ through the Mix operation formulated as follows:

$$a = \mathrm{Mix}(b, a^-, \hat{a}) = (1-b) \odot a^- + b \odot \hat{a}. \tag{4}$$

This procedure guarantees that the final decision $a$ replicates the same actions as the previous action $a^-$ in dimensions where *repetition* is selected, i.e. $(a - a^-) \odot (1 - b) = 0$. The detailed proof is straightforward and can be found in Appendix A.

Above all, the decision of the whole two-stage policy $\pi^{\mathrm{all}}$ can be described as follows:

$$\pi^{\mathrm{all}}(a|s, a^-) = \sum_b \beta(b|s, a^-) \int_{\hat{a}} \pi(\hat{a}|s, a^-, b) \cdot \delta\left(\mathrm{Mix}\left(b, a^-, \hat{a}\right) - a\right) \mathrm{d}\hat{a}. \tag{5}$$

### 4.2 POLICY EVALUATION

In this work, we employ two identical $Q$ functions to evaluate the performance of $\pi^{\mathrm{all}}$. Each $Q$ is trained utilizing Mean Square Error (MSE) loss with targets obtained based on the Bellman operator $\mathcal{T}$ formulated as follows:

$$\min \mathbb{E}_{(s,a) \sim \mathcal{D}} \left[Q(s, a) - \mathcal{T}Q(s, a)\right]^2, \text{ with } \mathcal{T}Q(s, a) = R(s, a) + \gamma \mathbb{E}_{P, \beta, \pi}\left[Q\left(s', a'\right)\right], \tag{6}$$

---

**Algorithm 1** Spatially Decoupled Action Repetition (SDAR) Algorithm

---
1: **Initialize:** Selection policy $\beta$, action policy $\pi$, buffer $\mathcal{D}$, critics $Q_{\{1,2\}}, Q_{\text{targ},\{1,2\}}$.
2: **for** each environment step $t$ **do**
3:     **if** a new episode starts **then**
4:         $b = [1, 1, ..., 1]_{|A|}$
5:     **else**
6:         $b \sim \beta(\cdot|s_t, a_{t-1})$
7:     **end if**
8:     Obtain $a_t$ through action policy, i.e. Eq. (2)-(4)
9:     Execute $a_t$ and observe $s_{t+1}$ with reward $r_t$.
10:     Store transition $(s_t, a_{t-1}, a_t, r_t, s_{t+1}, d_t)$ into buffer $\mathcal{D}$
11:     Sample a batch of data $\{(s, a^-, a, r, s', d)\}$ from $\mathcal{D}$
12:     Compute $a'$ for state $s'$ using $\beta$ and $\pi$.
13:     Update $Q_{\{1,2\}}$ through Eq. (6)
14:     **if** $t \mod \text{policy\_delay}$ **then**
15:         Update $\theta^\pi$ through Eq. (7)
16:         Update $\theta^\beta$ through Eq. (8) or Eq. (9)
17:         Update $\alpha_\beta$ and $\alpha_\pi$ through Eq. (10)
18:         Update target networks $Q_{\text{targ},\{1,2\}}$ based on soft updates.
19:     **end if**
20: **end for**

---

where $\mathcal{D}$ is the replay buffer. $\mathcal{T}$ in this work is similar to that in standard off-policy RL algorithms (Silver et al., 2014; Lillicrap, 2015), which is demonstrated to converge to the optimal $Q$ after sufficient iterations. In addition, we formulate two $Q$ functions based on neural networks, and train using clipped double-Q learning to alleviate the overestimation problem(Fujimoto et al., 2018). More details are given in Algorithm 1.

### 4.3 POLICY IMPROVEMENT

We propose to optimize the policy $\beta$ and $\pi$ through maximizing the objective $J(\theta^\beta, \theta^\pi)$ formulated as follows, where $\theta^\beta$ and $\theta^\pi$ are learnable parameters of $\beta$ and $\pi$ correspondingly.

$$J(\theta^\beta, \theta^\pi) = \underbrace{\mathbb{E}_{(s,a^-)\sim\mathcal{D}}\mathbb{E}_{b\sim\beta, \hat{a}\sim\pi}}_{\text{make decisions on samples}} \left[ \underbrace{Q(s,a)}_{\text{max } Q \text{ values}} \underbrace{-\alpha_\beta \log \beta\left(b|s, a^-\right) - \alpha_\pi \log \pi\left(\hat{a}|s, a^-, b\right)}_{\text{entropy-based exploration}} \right], \quad (7)$$

where entropy terms $\mathbb{E}\left[-\log \beta(b|s, a^-)\right]$ and $\mathbb{E}\left[-\log \pi(a|s, a^-, b)\right]$ are utilized to encourage exploration during training, which is widely utilized in prior works (Haarnoja et al., 2018; Yu et al., 2021). $\alpha_\beta$ and $\alpha_\pi$ are temperature parameters to adjust exploration strategy, both of which are tuned automatically during training. As illustrated in Sec. 4.1 and Sec. 4.2, the computation process of $\pi$ and $Q$ functions are both differentiable. Therefore, the action policy $\pi$ can be directly optimized using gradient descent with $\nabla_{\theta^\pi} J$.

In order to optimize selection policy $\beta$, we can transform objective $J$ into the following formulations:

$$\max_{\theta^\beta} \mathbb{E}_{(s,a^-)\sim\mathcal{D}} \sum_{b\in\mathcal{B}} \beta(b|s, a^-)\mathbb{E}_{\hat{a}\sim\pi}\left[Q(s,a) - \alpha_\beta \log \beta\left(b|s, a^-\right) - \alpha_\pi \log \pi\left(\hat{a}|s, a^-, b\right)\right], \quad (8)$$

where $\mathcal{B} = \{0,1\}^{|A|}$ is the action space for selection policy $\beta$. In contrast to Eq. (7), the new objective in Eq. (8) substitutes the expectation operator $\mathbb{E}_{b\sim\beta}$ with the summation operator $\sum_{b\in\mathcal{B}}$ by listing all possible situations $b \in \mathcal{B}$. The new objective is differentiable for $\theta^\beta$, thus can be used to optimize $\beta$ through gradient descent directly.

However, Eq. (8) is quite computationally expensive, because we are required to compute the score for all $b \in \mathcal{B}$, which needs to calculate $Q$ and $\pi$ for $|\mathcal{B}| = 2^{|A|}$ times. Thus, this method is only practical for tasks with small action spaces, such as LunarLander with $|A| = 2$. For tasks with large action spaces such as Humanoid ($|A| = 17$), the selection policy $\beta$ can be optimized by sampling

several $b \in \mathcal{B}$, which is formulated as follows:

$$\max_{\theta^\beta} \mathbb{E}_{\mathcal{D}} \mathbb{E}_{b \sim \beta_{\mathrm{old}}, \hat{a} \sim \pi} \left[ Q\left(s, a\right) - \alpha_\beta \log \beta_{\mathrm{old}} \left(b|s, a^-\right) - \alpha_\pi \log \pi \left(\hat{a}|s, a^-, b\right) \right] \cdot \frac{\beta\left(b|s, a^-\right)}{\beta_{\mathrm{old}}\left(b|s, a^-\right)},$$
(9)

where $\beta_{\mathrm{old}}$ denotes the old selection policy before optimization. This objective finds the expectation value $\mathbb{E}_{b \sim \beta}[\cdot]$ in Eq. (7) utilizing importance sampling. Compared to uniform sampling, sampling with $\beta_{\mathrm{old}}$ increases the probability of sampling $b$ with high $Q$ values during training, leading to higher sampling efficiency and training stability. As illustrated in Eq. (9), we are only required to compute $Q$ and $\pi$ given several $b$ sampled from $\beta$, thus is more computationally efficient and practical than Eq. (8) in applications.

In addition, temperature parameters $\alpha_\beta$ and $\alpha_\pi$ are tuned automatically following the objective:

$$\min_{\log(\alpha_\beta), \log \alpha_\pi} \mathbb{E}_{\beta, \pi} \left[ -\log\left(\alpha_\beta\right)\left(\log \beta(b|s, a^-) + \mathcal{H}_\beta\right) - \log \alpha_\pi \left(\log \pi(\hat{a}|s, a^-, b) + \mathcal{H}_\pi\right) \right], \quad (10)$$

where $\mathcal{H}_\beta$ and $\mathcal{H}_\pi$ are target entropies. During training, $\log \alpha_\beta$ and $\log \alpha_\pi$ are updated automatically through solving Eq. (10) with gradient descent, which is similar to prior works (Haarnoja et al., 2018). The detailed settings of the target entropies are described in Appendix B.1.

The whole training process of SDAR is illustrated in Algorithm 1, which iteratively collects data, trains critics, and updates policies. As illustrated in line 10, different to typical DRL algorithms, the previous action $a_{t-1}$ is stored in the buffer $\mathcal{D}$, which is utilized to improve $\theta^\pi$ and $\theta^\beta$ described in line 15-16. Besides, the agent is forced to choose *act* at the initial step of an episode, i.e. $b = [1, ..., 1]_{|A|}$, because $a^-$ is absent at this state.

## 5  EXPERIMENTS

### 5.1  EXPERIMENTAL SETUP

**Tasks:**  In this work, we conduct experiments on multiple continuous control tasks, which are categorized into the following three types of scenarios. More details are given in Appendix B.2.

(a) **Classic Control**: Several control tasks with small observation and action spaces, including *MountainCarContinuous*, *LunarLanderContinuous*, and *BipedalWalker*.

(b) **Locomotion**: Locomotion tasks based on the MuJoCo (Todorov et al., 2012) simulation environment: *Walker2d*, *Hopper*, *HalfCheetah*, *Humanoid*, and *Ant*.

(c) **Manipulation** tasks including *Pusher*, *Reacher*, and *FetchReach*.

**Baseline Methods.**  In this experiment, we compare the performance of SDAR with the following baseline methods: (a) **SAC**: vanilla Soft Actor Critic (Haarnoja et al., 2018) algorithm; (b) **N-Rep**: repeat policy actions for $n$ times, where $n$ is a *fixed* number (Mnih et al., 2015); (c) **TempoRL**: repeat policy actions dynamically in an *open-loop* manner, using a *skip-policy* to predict repetition steps (Biedenkapp et al., 2021); (d) **UTE**: Uncertainty-aware Temporal Extension (Lee et al., 2024) enhances training efficiency through incorporating uncertainty estimation of repeated action values into the *open-loop* repetition framework. (e) **TAAC**: Temporally Abstract Actor-Critic (Yu et al., 2021), which repeats actions dynamically in a *closed-loop* manner. TAAC determines whether to repeat the previous action at each step utilizing a switch policy. More details of the baseline methods are illustrated in Appendix B.3.

This study trains each method on various tasks using multiple random seeds over a range of 100K to 3M steps, depending on the complexity of the task. More settings including hyperparameters settings are described in Appendix B.1.

### 5.2  EXPERIMENT RESULTS ON SAMPLE EFFICIENCY

The training curves are illustrated in Fig. 3, and the AUC scores are shown in Table 1. The AUC scores are normalized into $[0, 1]$ to evaluate the sample efficiency across different tasks, where 0.0

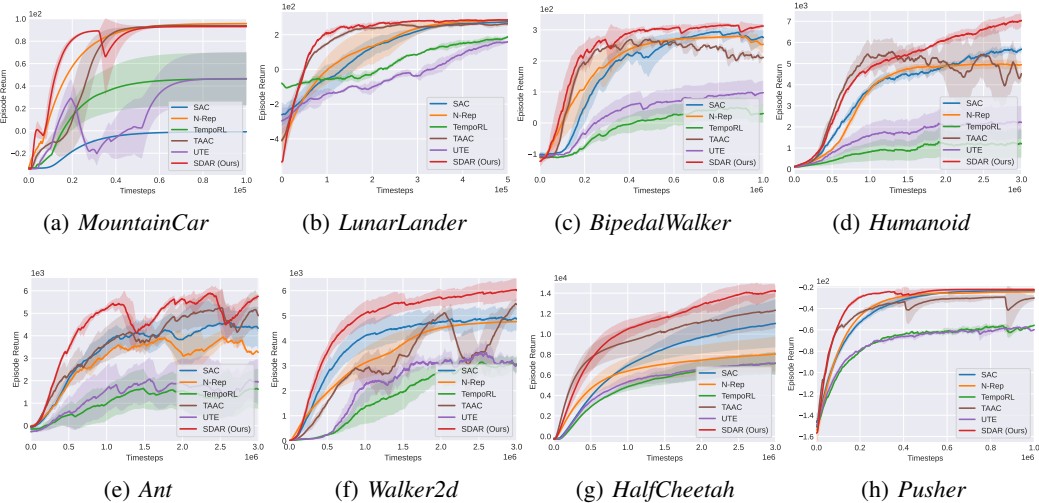

Figure 3: Learning curves of SDAR (red) in various tasks against baseline methods. Each method is trained with at least 10 random seeds. The lines denote the mean episode return, while shaded regions denote the standard error during training. As shown in the figures, our method generally achieves higher sample efficiency in various tasks compared to previous methods. More learning curves are given in Appendix. D.

Table 1: Experiment results on AUC scores (area under the learning curves) in different types of environments, where higher AUC scores indicate higher sample efficiency of corresponding DRL algorithms. The average scores along with the standard errors are calculated across various tasks within the respective category. The results are normalized to evaluate performance across different tasks, where 1.0 denotes the best performance. The best results are boldfaced.

| Env. Category | Normalized AUC Score | | | | | |
|---|---|---|---|---|---|---|
| | SAC | N-Rep | TempoRL | UTE | TAAC | SDAR |
| Classic Control | 0.60±0.13 | 0.89±0.01 | 0.66±0.02 | 0.55±0.03 | 0.92±3E-3 | **1.0±0.0** |
| Locomotion | 0.78±2E-3 | 0.71±7E-3 | 0.35±0.02 | 0.43±0.01 | 0.80±0.02 | **1.0±0.0** |
| Manipulation | 0.91±4E-5 | 0.90±8E-4 | 0.77±0.02 | 0.79±0.02 | 0.95±6E-3 | **1.0±0.0** |
| **Average** | 0.76±0.02 | 0.83±0.01 | 0.59±0.05 | 0.59±0.03 | 0.90±6E-3 | **1.0±0.0** |

denotes the performance of random policies without effective training, while 1.0 denotes the performance of the best method. See Appendix C for the detailed computation process of the normalization. More experiment results are given in the Appendix D.

As illustrated in Fig. 3, our method **SDAR** (red lines) achieves higher sample efficiency than other baseline methods in various continuous control environments, corresponding to higher AUC scores shown in Table 1. This demonstrate effectiveness of performing *act-or-repeat* selection for each action dimension individually. Such spatially decoupled repetition framework enhances action persistence while maintaining policy flexibility, leading to efficient exploration and training. In the following, we analyze the performance of each method individually:

- Naive action repetition **N-Rep** achieves unstable improvement compared to vanilla **SAC**, but underperforms closed-loop repetition approaches generally. As described in Table 1, **N-Rep** achieves higher AUC scores than **SAC** in classic control tasks, such as *MountainCar* shown in Fig. 3(a). Naive repetition improves the persistence of actions during exploration, accelerating the training progress effectively in simple tasks. Nevertheless, this approach lacks flexibility, which

could negatively impact the performance in tasks requiring agile movements, such as *HalfCheetah* (Fig. 3(g)) in locomotion domains.

- Open-loop methods (**TempoRL** and **UTE**) show minor improvements over **SAC** in classic control tasks, while performing worse performance in other domains. These approaches force the agent to repeat actions for a predicted number of steps, eliminating the possibility of early termination. This inflexibility makes such methods difficult to be utilized in tasks requiring agile movements, such as *Humanoid* shown in Fig. 3(d).

- The closed-loop method **TAAC** outperforms **SAC** and open-loop approaches generally in various classes of tasks. **TAAC** introduces a switch policy to check whether to repeat previous actions at each step, which solves the lack of a termination mechanism in open-loop repetition methods. This leads to a more flexible repetition, which is suitable for both simple tasks and locomotion tasks requiring agile motions, such as *Ant* shown in Fig. 3(e).

  However, **TAAC** treats all action dimensions as a whole during *act-or-repeat* selection, which downgrades the effectiveness of action repetition. Take *Humanoid* and *HalfCheetah* shown in Fig. 3(d) and Fig. 3(g) respectively as examples, **TAAC** (brown lines) demonstrates superior sample efficiency during the initial training phase, specifically between 0 and 1M steps. However, in later stages, the agent's learning speed diminishes compared to **SDAR**, which even results in performance decline in the *Humanoid* task depicted in Fig. 3(d). This is because the *Humanoid* is composed of multiple actuators, where some actuators are required to select *act* for action diversity, while others choosing *repeat* for action persistence at the same time. However, **TAAC** is constrained to repeat actions of all actuators simultaneously, leading to either inadequate repetition for inefficient exploration, or excessive repetition to damage the policy performance.

- Our method **SDAR** outperforms other methods in various tasks, including **TAAC** in *locomotion* tasks. **SDAR** decouples action dimensions during closed-loop *act-or-repeat* selection, which is more flexible and suitable for various types of tasks. Taking the *Humanoid* task illustrated in Fig. 3(d) as an instance, **SDAR** compromises marginal efficiency in the initial stages to achieve greater overall efficiency and stability throughout the entire training process. Above all, **SDAR** achieves a better balance between action persistence for efficient exploration, and action diversity for high policy performance.

## 5.3 POLICY PERFORMANCE AND FLUCTUATION

In this work, we assess the policies trained through each method on the following metrics: **(1) Episode Return**: the average cumulative reward acquired by the policy in an episode, assessing the policy's capability to solve the task successfully. **(2) Action Persistence Rate (APR)** and **(3) Action Fluctuation Rate (AFR)** formulated as follows:

$$
\text{APR} = \frac{1}{1-p}, \; p = \mathbb{E}_\pi \left[ \frac{1}{T \cdot |A|} \sum_{t=1}^{T} \sum_{i=1}^{|A|} \delta \left( a_t^{(i)} - a_{t-1}^{(i)} \right) \right]; \; \text{AFR} = \mathbb{E}_\pi \left[ \frac{1}{T} \sum_{t=1}^{T} \| a_t - a_{t-1} \| \right],
$$

where $T$ denotes the trajectory length, $a_t^{(i)}$ denotes the action at the $t$-th step in the $i$-th dimension, and $p$ denotes the average repetition probability of $a_t^{(i)}$ at each step. In this work, APR represents the average interval that two new decisions are made. APR evaluates *action persistenc* of the policy, where a larger APR denotes more repetition times during interactions. AFR evaluates the mean amplitude of fluctuations between successive steps, where a smaller AFR indicates smoother actions with fewer fluctuations.

The results of episode return, APR, and AFR are shown in Table 2. As illustrated in the tables, although the baseline method such as **N-Rep** achieves excellent results in both APR and AFR, it suffers a considerable reduction on policy effectiveness, corresponding to lower episode return compared to **SAC**. This is unacceptable and makes such methods impractical for real-world applications. In contrast, our method **SDAR** outperforms baseline methods on episode return, while achieving a higher APR and a lower AFR than the vanilla DRL, demonstrating the effectiveness of our method. In addition, we can also observe that:

- Naive repetition method **N-Rep** achieves high APR and low AFR, at the expense of performance reduction on episode returns. This is because the actions in **N-Rep** are repeated for frozen times,

Table 2: Experiment results on **Episode Return**, **APR (higher is better)**, and **AFR (lower is better)** in different tasks. The best results on episode returns are boldfaced. Our method SDAR outperforms baseline methods on episode returns generally, achieving excellent action persistence with fewer fluctuation than vanilla methods.

| Tasks | Episode Return (Mean ± Standard Error) | | | | | |
| --- | --- | --- | --- | --- | --- | --- |
| | Action Persistence Rate (APR) / Action Fluctuation Rate (AFR) | | | | | |
| | SAC | N-Rep | TempoRL | UTE | TAAC | SDAR (Ours) |
| LunarLander | 275.9±6.83 | 280.8±1.21 | 281.5±7.22 | **282.8±8.74** | 261.4±18.9 | 282.2±5.84 |
| | 1.00 / 0.09 | 4.00 / 0.08 | 1.43 / 0.18 | **1.38 / 0.36** | 3.05 / 0.11 | 11.18 / 0.10 |
| Walker2d | 5305±367 | 4724±163 | 2866±897 | 2986±836 | 5660±394 | **6028±406** |
| | 1.00 / 0.15 | 4.00 / 0.09 | 5.74 / 0.26 | 7.81 / 0.24 | 1.30 / 0.22 | **2.96 / 0.12** |
| HalfChee. | 13122±2877 | 8378±1753 | 8065±1799 | 7917±293 | 11148±3921 | **15131±1279** |
| | 1.00 / 0.68 | 2.00 / 0.47 | 2.10 / 0.66 | 2.69 / 0.58 | 1.02 / 0.61 | **1.22 / 0.62** |
| Humanoid | 6184±717 | 5074±310 | 1022±397 | 2595±334 | 7308±244 | **7483±288** |
| | 1.00 / 0.28 | 4.00 / 0.09 | 5.27 / 0.15 | 5.69 / 0.18 | 1.21 / 0.25 | **1.67 / 0.19** |
| Pusher | -22.5±1.40 | **-21.2±1.08** | -48.2±2.05 | -41.3±5.56 | -30.5±1.85 | -21.3±1.26 |
| | 1.00 / 0.022 | **4.00 / 0.019** | 1.15 / 0.032 | 1.01 / 0.018 | 1.75 / 0.031 | 1.69 / 0.015 |
| **Average** | 0.89±0.07 | 0.81±0.19 | 0.59±0.33 | 0.64±0.27 | 0.91±0.10 | **1.00±0.001** |
| | 1.00 / 0.245 | 3.60 / 0.150 | 3.12 / 0.257 | 3.71 / 0.276 | 1.66 / 0.244 | **3.75 / 0.208** |

where repeated actions may be unreasonable in some states. This inflexible repetition strategy cannot be agilely adjusted during interaction, leading to performance reduction, especially in *locomotion* tasks such as *HalfCheetah*.

- Open-loop methods achieve high APR, indicating a lot of action repetitions during interaction, at the expense of more fluctuations and superior episode return performance. For instance, **UTE** achieves higher APR (3.71) than **SAC** (1.0), with large fluctuation and low episode return. This suggests that *the increment of repetition times in an inflexible manner may be harmful to policy effectiveness and smoothness*. In contrast, **TAAC** solves this problem, achieving a moderate AFR (1.66), with a comparable AFR (0.244) and high episode return (0.91).

- Different to **UTE** and **SDAR**, although **SDAR** also selects a lot of *repeat* during interaction with a high APR (3.69), **SDAR** achieves excellent performance on fluctuation and episode returns simultaneously. This is because **SDAR** performs repetitions in a more flexible manner. The decoupling design is consistent with the requirement of continuous control tasks, thus can conduct repetition without harms to episode returns and fluctuation performance.

## 5.4 VISUALIZATION OF ACT-OR-REPEAT SELECTION

In order to analyze the difference between repetition behaviors of **SDAR** and previous methods, we perform visualization of *act-or-repeat* selections in *LunarLander* and *Walker2d*. The results are illustrated in Fig. 4(a) and Fig. 4(b), where dark blue and light blue blocks denote *act* and *repeat* respectively.

As shown in the figures, different dimensions require different decision frequencies, corresponding to actions being repeated at distinct steps for each dimension. Take *LunarLander* in Fig. 4(a) for example, the lateral boosters ($a_1$) change actions frequently to adjust the rocket pose, while the main engine ($a_0$) adjusts decisions occasionally.

Table 3: Experiment results of each action dimensions in *Walker2d*.

| APR Val. | TAAC | SDAR |
| --- | --- | --- |
| Thigh ($a_0$) | 1.30 | **3.70** |
| Leg ($a_1$) | 1.30 | **2.17** |
| Foot ($a_2$) | 1.30 | **3.44** |
| Episode Ret. | 5660 | **6028** |
| AFR | 0.22 | **0.12** |

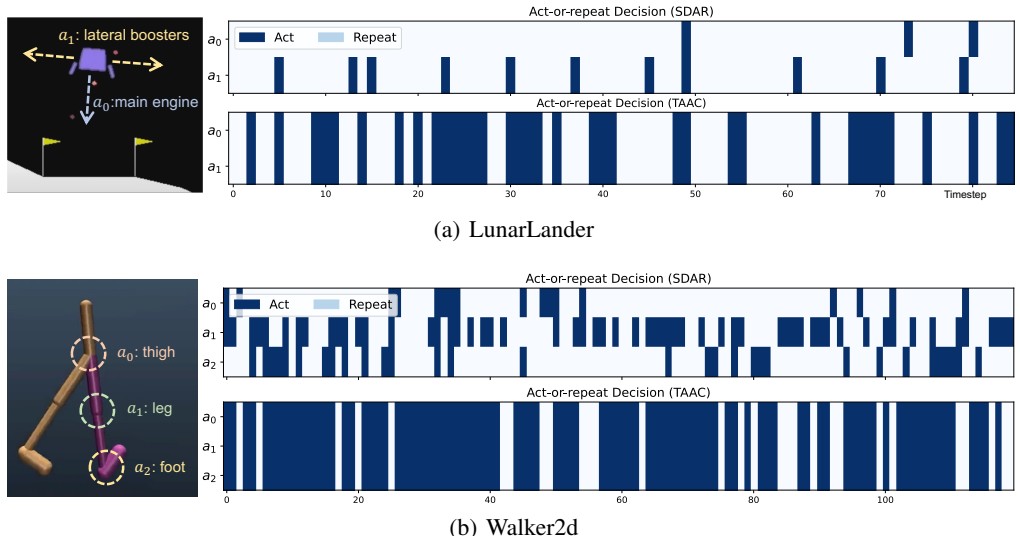

(a) LunarLander

(b) Walker2d

Figure 4: Visualization of *act-or-repeat* selections of **SDAR** and **TAAC** algorithms in *LunarLander* and *Walker2d* tasks. The $x$-axis denotes timesteps, and the $y$-axis denotes different action dimensions. The light blue blocks indicate *repetition*, while dark blue blocks represent *act*, i.e. change actions in the corresponding action dimensions.

**SDAR** can adjust action persistence for each dimension automatically through spatially decoupled framework, offering more flexible repetition strategies compared to **TAAC**. As described in Fig. 4(b) and Table 3, **SDAR** controls the walker with higher persistence for the leg joints, while lower persistence for thighs and feet. In contrast, all joints in **TAAC** are required to repeat simultaneously, resulting in same persistence for each joint with lower repetitions behaviors.

**SDAR** achieves high action persistence, while maintaining agility required by the task, leading to higher policy performance with fewer action fluctuations. As shown in Table 3, **SDAR** obtains higher APR than **TAAC** in *Walker2d*, while achieving higher episode return with lower AFR, demonstrating the effectiveness of our framework in continuous control tasks.

## 6 CONCLUSION

In this work, we propose a novel action repetition framework for continuous control tasks called SDAR, which implements closed-loop *act-or-repeat* selection for each action dimension individually. Such a spatially decoupled design improves the flexibility of repetition strategies, leading to improved balance between action *persistence* and *diversity*, while maintaining action agility for continuous control tasks. Experiments are conducted in various task scenarios, where SDAR achieves higher sample efficiency, superior policy performance, and reduced action fluctuation, demonstrating the effectiveness of our method.

This work provides insights into spatially decoupled framework for action repetition and temporal abstraction. In this study, the selection policy $\beta$ is designed to decide *act-or-repeat* for each dimension independently, without accounting for inter-dimensional correlations. For instance, actuators in humanoid robots can be divided into multiple categories, with each category typically requiring same decisions during *act-or-repeat* selections. This issue will be further researched in future works.

ACKNOWLEDGMENTS

This work was supported by the National Natural Science Foundation of China (Grant No. 62373242 and No.92248303), Shanghai Municipal Science and Technology Major Project (Grant No. 2021SHZDZX0102).

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

## A  PROOF OF THE STATEMENT IN SECTION 4.1

**Statement.**  Given $\forall s \in \mathcal{S}, a^- \in \mathcal{A}, b \in \{0,1\}^{|A|}$, based on the two-stage policy described in Sec. 4.1, the output action $a$ replicates the same actions as $a^-$ in *repetition* dimensions $\{i | b_i = 0, 1 \leq i \leq |A|\}$.

*Proof.*  As described in Sec. 4.1, $a = (1 - b) \odot a^- + b \odot \hat{a}$. Thus, for $\forall b, a^-$, we can obtain that:

$$
\begin{aligned}
(a - a^-) \odot (1 - b) &= \left((1 - b) \odot a^- + b \odot \hat{a} - a^-\right) \odot (1 - b) \\
&= b \odot \left(\hat{a} - a^-\right) \odot (1 - b) \\
&= b \odot (1 - b) \odot \left(\hat{a} - a^-\right) \\
&= \mathbf{0}
\end{aligned}
\tag{11}
$$

For $\forall i \in \{i | b_i = 0\}$, we have $[(a - a^-) \odot (1 - b)]_i = a^{(i)} - a^{-,(i)} = 0$, where $a^{(i)}$ denotes the action in the $i$-th dimension. Thus, $a$ replicates the same actions as $a^-$ in dimensions choosing *repetition*.

$\square$

## B  EXPERIMENT DETAILS

### B.1  ALGORITHM SETTINGS

Table 4: Hyper-parameter settings for SDAR algorithm.

| Parameter | Setting | Parameter | Setting |
|-----------|---------|-----------|---------|
| Learning rate ($\pi$) | $3 \times 10^{-4}$ | Learning rate ($\beta$) | $3 \times 10^{-4}$ |
| Learning rate ($Q$) | $1 \times 10^{-3}$ | Learning rate ($\alpha$) | $1 \times 10^{-3}$ |
| Optimizer | Adam | Discount factor $\gamma$ | 0.99 |
| Batch size | 256 | Policy delay | 2 |
| Soft update $\tau$ | 0.005 | Sample number ($b$) | 10 |

The hyper-parameter settings ar shown in Table. 4. In addition, we need to tune the target entropies $\mathcal{H}_\beta$ and $\mathcal{H}_\pi$ to improve the efficiency of the entropy-based exploration described in Eq. (10). In this work, we utilize $\mathcal{H}_\pi = -|A|$, where $|A|$ denotes the size of the action space, corresponding to the recommendation given in (Haarnoja et al., 2018). Besides, $\mathcal{H}_\beta = \lambda \cdot |A| \log 2$, where $\lambda$ is a hyper-parameter tuned according to the tasks. In this work, we set $\lambda \in [0.4, 0.6]$, which achieves excellent performance on sample efficiency and policy effectiveness.

As illustrated in Sec. 4.1, SDAR is composed of two policy networks $\beta$ and $\pi$ based on MLP. In this work, we design two MLPs with the same structures: $(|S| + |A| \rightarrow 256 \rightarrow 256 \rightarrow |A|)$ with ReLU activation functions.

**SDAR** policies for tasks with $|A| \leq 3$ are trained through Eq. (8), such as *Reacher* and *LunarLanderContinuous*. Policies in other environments such as *HalfCheetah* and *Humanoid* are optimized through Eq. (9) for lower computation costs.

### B.2  EXPERIMENT TASKS

The experiment tasks are listed in Table 5, where the action spaces of all tasks are normalized as $[-1, 1]$ for convenience. All tasks are constructed based on Gymnasium (Plappert et al., 2018). The *FetchPickandPlace* and *FetchReach* tasks are implemented by Gymnasium-Robotics (Plappert et al., 2018), where *observation* and *desired_goal* are concatenated as the observation in this experiment.

Table 5: Descriptions of experiment tasks in this work.

| Caterogy | Task Name | Obervation Space | Action Space |
|---|---|---|---|
| Classic Control | MountainCarcontinuous | $\mathbb{R}^2$ | $[-1, 1]^1$ |
| | LunarLanderContinuous | $\mathbb{R}^8$ | $[-1, 1]^2$ |
| | BipedalWalker | $\mathbb{R}^{24}$ | $[-1, 1]^6$ |
| Locomotion | HalfCheetah | $\mathbb{R}^{17}$ | $[-1, 1]^6$ |
| | Humanoid | $\mathbb{R}^{376}$ | $[-1, 1]^{17}$ |
| | Walker2d | $\mathbb{R}^{17}$ | $[-1, 1]^6$ |
| | Hopper | $\mathbb{R}^{11}$ | $[-1, 1]^3$ |
| | Ant | $\mathbb{R}^{27}$ | $[-1, 1]^8$ |
| Manipulation | FetchReach | $\mathbb{R}^{13}$ | $[-1, 1]^4$ |
| | Pusher | $\mathbb{R}^{23}$ | $[-1, 1]^7$ |
| | Reacher | $\mathbb{R}^{11}$ | $[-1, 1]^2$ |

### B.3 BASELINES

(1) **SAC** (Haarnoja et al., 2018) is a famous model-free RL in continuous control domains, which trains policies efficiently with entropy-based exploration strategies. In this work, we utilize SAC implementation and hyper-parameter settings proposed in CleanRL (Huang et al., 2022)[1].

(2) **N-Rep** forces the agent to repeat the actions output by the policy for $n$ times, where $n$ is a hyper-parameter. In this work, this algorithm is implemented based on SAC, where $n$ is tuned in $[2, 5]$ to achieve the balance between persistence of action and diversity.

(3) **TempoRL** (Biedenkapp et al., 2021) repeats policy actions dynamically in an *open-loop* manner, using a *skip-policy* to predict when to make the next decision. In this work, this method is implemented based on the official repository[2].

(4) **UTE** (Lee et al., 2024) is improved based on **TempoRL**, which performs repetition while incorporating the estimated uncertainty of the repeated action values. In this work, this method is implemented based on the official repository [3].

(4) **TAAC** (Yu et al., 2021) is a closed-loop repetition framework, which determines whether to repeat the previous action at each step utilizing a switch policy. In this work, this experiment is conducted utilizing the official implemenatation[4].

### B.4 COMPUTATION RESOURCES

In this work, we conduct all experiment utilizing NVIDIA RTX 3090 GPU and Pytorch 2.1 with CUDA 12.2. The training time of our method compared to vanilla SAC is shown in Table 6.

As shown in Table 6, our method requires less training times than vanilla **SAC** to achieve the same policy performance, because of the higher sample efficiency of **SDAR**, requiring less training steps in this experiment. In addition, there exist two policy networks $\beta$ and $\pi$ in the SDAR, both of which need to be optimized through gradient descent. However, $\beta$ and $\pi$ can be optimized in parallel during training, achieving an improved time efficiency, which will be researched in future works.

## C NORMALIZATION OF PERFORMANCE

Different tasks can have vastly different reward scales, thus cannot be directly averaged. In order to depict the average episode return and AUC scores across various tasks, we compute the normalized normalized score (n-score) of each method, which is widely utilized in prior works Hessel et al.

---

[1] https://github.com/vwxyzjn/cleanrl

[2] https://github.com/automl/TempoRL

[3] https://github.com/oh-lab/UTE-Uncertainty-aware-Temporal-Extension-/

[4] https://github.com/HorizonRobotics/alf

Table 6: The computational cost for the training of SAC and SDAR algorithms. Both two methods are trained to achieve the same performance as SAC with sufficient steps.

| Tasks | Steps Cost | | Time Cost | |
|---|---|---|---|---|
| | SAC | SDAR | SAC | SDAR |
| LunarLander | 300K | 150K | 60min | 36min |
| BipedalWalker | 600K | 300K | 108min | 63min |
| Walker2d | 1.5M | 700K | 282min | 167min |
| Humanoid | 3.0M | 1.5M | 595min | 365min |

(2017); Yu et al. (2021). Take normalized episode return as an example, given an episode return $Z$, its n-score is calculated as $Z_{norm} = \frac{Z - Z_0}{Z_1 - Z_0}$, where $Z_0$ and $Z_1$ denote the episode return of random policies and vanilla DRL policies respectively. The n-score $Z_{norm} \in [0, 1]$, thus can be averaged across different tasks.

## D  ADDITIONAL TRAINING CURVES

Additional training curves of each method in additional tasks are illustrated in Fig. 5.

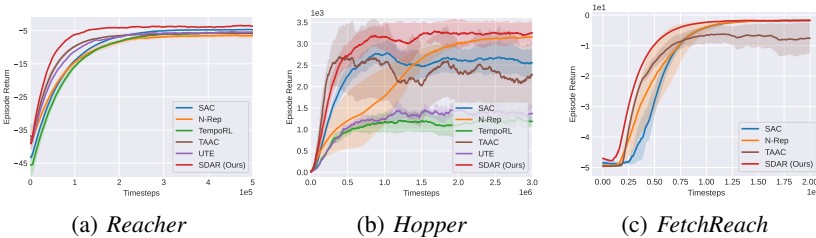

(a) *Reacher*  (b) *Hopper*  (c) *FetchReach*

Figure 5: Learning curves of SDAR (red) in additional tasks against baseline methods.

