# OpenReview forum: "Select before Act: Spatially Decoupled Action Repetition for Continuous Control"
_ICLR.cc/2025/Conference — ICLR 2025 Poster_

### Official Review · Reviewer_sXke · 2024-11-03

**Soundness:** 2
**Presentation:** 3
**Contribution:** 2
**Rating:** 5
**Confidence:** 4

**Summary:**

The paper proposes a novel reinforcement learning algorithm, "Spatially Decoupled Action Repetition", which decouples policy and repetition. This philosophy is achieved by learning two policies, selection and action policies. The selection policy outputs which action dimension should be masked and the action policy proposes a new action. The proposed algorithm was evaluated on several control domains, such as classical control, locomotion, manipulation.

**Strengths:**

* The paper demonstrates a novel algorithm that decouples repetition and control, which is an interesting idea.
* The proposed algorithm performs well on a wide range of control domains.

**Weaknesses:**

I don't think I am fully convinced by the paper.
1. Overall, I don't think I understand why the proposed decoupling is important. It is an interesting idea that extends the existing action dimension. But why? In continuous control, I believe many challenging problems mandate new actions at every single time step. Even many discrete action problems may require the same.
2. In other words, what will be the problem class that must require decoupled action repetition?
3. As a result, I currently think there can be other reasons that the proposed method outperforms the baselines, such as better hyperparameters or larger network sizes. Please note that I am not questioning the integrity of the proposed authors; there can be other reasons that we may not understand.

**Questions:**

* Would you be able to give us the intuition as to why the proposed method "should" work better than the baselines? Examples or toy-problems would be appreciated.
* The performance is only 10% better than SAC - many other ensemble-based approaches, such as RED-Q or Dro-Q, might perform much better than the proposed method.

---

> ### Author Response · Authors · 2024-11-16
> **Response to Reviewer #sXke (Part 1)**
>
> Thanks for your detailed review and helpful comments. We hope that our answers will help to resolve your concerns.
>
> ## Weakness:
>
> > 1. Overall, I don't think I understand why the proposed decoupling is important. It is an interesting idea that extends the existing action dimension. But why?
> >
> >    In continuous control, I believe many challenging problems mandate new actions at every single time step. Even many discrete action problems may require the same.
>
> This work focuses on improving action persistence, which has been widely discussed and demonstrated effective in previous works, even in challenging tasks such as locomotion scenarios [1-5].
>
> We will analyze explain the effectiveness of SDAR  through the following subquestions:
>
> 1. **Why SDAR achieves higher sample efficiency than vanilla DRL and previous repetition methods ?**
>
> - Previous RL studies have pointed out the significant influence of action persistence on training efficiency of RL methods [1-6]. It is important to take proper trade-off between action persistence and diversity during training[2].
> - Previous repetition works aim to improve action persistence in a flexible manner: Repeat for given steps (frame-skipping / N-rep in this work) -> Open-loop repetitions (TempoRL, UTE) -> Closed-loop methods (TAAC)
> - SDAR achieves a more flexible repetition strategy in spatially decoupled manner. It improves the balance between persistence and diversity.
>
> 2. **Why improving action persistence is effective for common task scenarios?**
>
> There exist multiple reasons for this phenomenon:
>
> - Improving action persistence leads to deeper exploration with faster training efficiency in common decision tasks [1-6], Commonly practical tasks prefer policies to output similar actions in most successive steps, such as path planning and robot manipulation.
> - In MDP, the decision frequency is fixed, which is difficult to be set exactly as the minimum value that doesn't comprise optimal control while leaving no room for temporal abstraction[4].  Besides, the optimal control frequency may vary during the whole episode, leaving room for action repetitions.
> - In tasks with |A|>1, each action dimension commonly requires different persistence with distinct control frequencies [5], such as robot joint controllers operating at different frequencies.
>
> Above all, (1) SDAR achieves improved balance between persistence and diversity compared to previous methods. (2) Flexible persistence is important for continuous control policies in common scenarios.
>
>
>
> [1] Dabney, Will, et al., Temporally-Extended epsilon-Greedy Exploration, ICLR 2021
>
> [2] Yu, Haonan, et al., Taac: Temporally abstract actor-critic for continuous control, NeurIPS 2021.
>
> [3] Metelli, Alberto Maria, et al., Control Frequency Adaptation via Action Persistence in Batch Reinforcement Learning, ICML 2020.
>
> [4] Grigsby, Jake, et al., Towards automatic actor-critic solutions to continuous control, NeurIPS 2021.
>
> [5] Lee, Jongmin, et al., Reinforcement learning for control with multiple frequencies, NeurIPS 2020.
>
> [6] Biedenkapp, André, et al.,TempoRL: Learning when to act, ICML 2021

---

> ### Author Response · Authors · 2024-11-16
> **Response to Reviewer#sXke (Part 2)**
>
> > 2. In other words, what will be the problem class that must require decoupled action repetition?
>
> As described in response to weakness 1, SDAR is suitable for common continuous control tasks, because flexible action persistence is commonly needed.
>
> The following are some features of tasks suitable for SDAR. Note that **only one requirement is needed to meet**.
>
> - The MDP of the task leaves room for action repetitions, such as
>
>   - The decision frequency of MDP is set larger (even a bit larger) than the minimum value required by the task.
>
>   - The minimum control frequency varies during interactions.
>
>   - Each action dimension requires different persistence with distinct minimum control frequencies.
>
> - The task prefers the policy to output similar actions in most successive steps. This makes exploration with action persistence deeper and faster.
>
> Above all, common tasks and MDP modeling in practical scenarios are suitable for SDAR in this work, such as path planning, robot locomotion, manipulation tasks, autonomous driving, etc.
>
>
> > 3. As a result, I currently think there can be other reasons that the proposed method outperforms the baselines, such as better hyperparameters or larger network sizes.
> >
> >    Please note that I am not questioning the integrity of the proposed authors; there can be other reasons that we may not understand.
>
> As described in response to weakness 1&2, SDAR outperforms previous methods because SDAR achieves improved trade-off between action persistence and action diversity through a more flexible repetition manner.
>
> In order to evaluate the performance of each method fairly, we try to avoid the influence of hyperparameters tuning in the experiment:
>
> - SDAR shares same hyper-parameters with SAC, including learning rate, hidden layer structures of networks, buffer sizes, soft update coefficient, etc.
> - For other methods such as TAAC, we utilize parameters recommended by authors. Besides, we also tried to tune these parameters such as network sizes and soft update  coefficient, which cannot achieve higher performance than official setups.

---

> ### Author Response · Authors · 2024-11-16
> **Response to Reviewer#sXke (Part 3)**
>
> ## Questions:
>
> > 1. Would you be able to give us the intuition as to why the proposed method "should" work better than the baselines? Examples or toy-problems would be appreciated.
>
> In response to weakness 1&2, we have explained the effectiveness intuitively and theoretically. In the following, we analyze the method with some examples.
>
> In Fig. 4, we visualize repetition schemas in *LunarLander* and *Walker2d* tasks, where x-axis denotes time steps, y-axis denotes action dimensions, *light-blue* / *dark-blue* denote *repetition* / *action* correspondingly.
>
> Fig. 4 (a) *LunarLander*:
>
> - SDAR: the lateral boosters ($a_1$) need to change actions frequently to adjust the rocket pose, while the main engine ($a_0$) only adjusts decisions occasionally at the later stage of the episode. This reduces times to make new decisions effectively. Persistence at each dimension can be adjusted flexibly to speed up exploration (Fig. 3 (b)) without cost on final performance.
> - In contrast, TAAC requires lateral boosters and the main engine to *act-or-repeat* simultaneously, which is inflexible leading to lower persistence and efficiency performance.
>
> Fig. 4 (b) *Walker2d*:
>
> - The thigh, leg, and foot joints are controlled with different persistence. The leg joint changes actions more frequently. This can be handled with SDAR properly. In contrast, TAAC implements less persistence with sacrifice on action diversity, leading to lower efficiency and final performance.
> - SDAR achieves higher persistence (2.96) than TAAC (1.30), with lower fluctuations (0.12) than TAAC (0.22). SDAR is trained faster (AUC 1.0) than TAAC (0.62), with better reward (6028) than TAAC (5660);
>
> > 2. The performance is only 10% better than SAC - many other ensemble-based approaches, such as RED-Q or Dro-Q, might perform much better than the proposed method.
>
> - **The main motivation of SDAR** is to achieve more flexible repetitions in the temporal abstraction field [7,8]. Previous works in this field focus on the temporal aggregation. In this work, we contribute a novel viewpoint and introduce the spatial design to take consideration of difference between action dimensions.  Methods such as RED-Q or Dro-Q improve sample efficiency through ensemble-based approaches, which are not contradictory with our method. Thus, our method can be incorporated into these ensemble-based approaches for a higher sample efficiency.
>
> - **The efficiency improvement:** As shown in Table 1, compared to SAC, SDAR achieves 31.5% improvement on sample efficiency,  with more action persistence (3.75) than SAC (1.0). Our method achieves 12.3% improvement on episode reward, because both SAC and SDAR are trained for sufficient steps until convergence. Therefore, the improvement is not essential.
>
> Thank you for your valuable feedback. This point is a potential research direction for future works.
>
>
>
> [7] Precup, Doina. Temporal abstraction in reinforcement learning. University of Massachusetts Amherst, 2000.
>
> [8] Sutton, Richard S., Doina Precup, and Satinder Singh. "Between MDPs and semi-MDPs: A framework for temporal abstraction in reinforcement learning." Artificial intelligence 1999.

---

> ### Author Response · Authors · 2024-11-25
>
> Dear Reviewer sXke,
>
> Thank you again for your time and efforts in reviewing this paper.
>
> Since the rebuttal period is ending soon, we would like to know if your concerns and questions have been addressed.  We would appreciate the opportunities to address any additional concerns you may have before the discussion phase ends.

---

> > ### Comment · Reviewer_sXke · 2024-11-29
> >
> > Thank you for your response.
> >
> > I still believe the proposed method relies on certain assumptions. For example, consider the task of tracking a sinusoidal joint trajectory. Repeating actions would inherently lead to suboptimal behavior due to the nature of the problem. Because we consider a single joint, it will be true regardless of whether the proposed method allows repetition at the per-joint level or not.
> >
> > With that in mind, I am more interested in understanding the properties that favor the proposed method. For instance, it seems that LunarLander falls into this category. How can we characterize such tasks? Is it possible to quantify these properties?

---

> > > ### Author Response · Authors · 2024-12-01
> > > **Response to Reviewer #sXke**
> > >
> > > Thank you for your valuable response. We agree that the task properties have influence on the advantage of (Spatially Decoupled) Action Repetition methods compared to naive DRL. Besides, the results in this work and previous works have demonstrated the effectiveness of (Spatially Decoupled) repetition-based  framework in common DRL continuous control tasks [1-6]. Each RL method is suitable for some tasks while maybe invalid in other scenarios. In the following, we try to clarify this issue based on the experiments and analysis.
> > >
> > > **Experiment and Analysis**
> > >
> > > According to your suggestion, the experiment of tracking a sinusoidal trajectory is conducted. Policies are trained to track three sinusoidal trajectories $[\phi(t,0), \phi(t,1), \phi(t,2)]$, where $\phi(t,i)=\sin(\frac{\pi}{2}t + \frac{\pi}{4}i)$ denotes the target value at time $t$ and the $i$-th dimension.  Policies are required to track 200 steps with decision interval 0.02 , where the policy needs to output new actions at each step.  The policy obtains reward $r = \frac{1}{3}\sum_i e^{-|a-\phi(t,i)|}$ at every step, where the optimal action $a^*_i=\phi(t,i)$ obtains the highest reward.
> > >
> > > The results of SAC and SDAR are shown as follows.
> > >
> > > | Timestep | SAC   | SDAR (Ours) |
> > > | -------- | ----- | ----------- |
> > > | 5k       | 155.6 | 165.4       |
> > > | 10k      | 180.3 | 187.2       |
> > > | 20k      | 188.6 | 192.8       |
> > > | 30k      | 192.2 | 195.6       |
> > > | 40k      | 194.1 | 195.7       |
> > > | 50k      | 195.1 | 196.1       |
> > >
> > > As shown in the table, both two methods solve the task with similar final performance, where SDAR achieves a bit higher sample efficiency. The analysis is described as follows:
> > >
> > > (1) SDAR explores the environment using policies with temporal persistence, which can achieve deeper exploration with higher training efficiency during training, even in this task that repeated actions are not preferred [1-3].
> > >
> > > (2) SDAR can adjust the repetition rate automatically to achieve higher rewards. In tasks that prefer low or even zero repetition rate, our methods can still achieve comparable final results, which is achieved by the flexible repetition schemas.
> > >
> > >
> > >
> > > [1] Dabney et al., Temporally-Extended epsilon-Greedy Exploration, ICLR 2021
> > >
> > > [2] Yu et al., Taac: Temporally abstract actor-critic for continuous control, NeurIPS 2021.
> > >
> > > [3] Grigsby et al., Towards automatic actor-critic solutions to continuous control, NeurIPS 2021.
> > >
> > > [4] Biedenkapp et al. TempoRL: Learning when to act. ICML 2021
> > >
> > > [5] Lee, Jongmin, et al., Reinforcement learning for control with multiple frequencies, NeurIPS 2020.
> > >
> > > [6] Karimi et al. Dynamic Decision Frequency with Continuous Options. IROS 2023

---

> > > ### Author Response · Authors · 2024-12-01
> > > **Official Comment by Reviewer sXke (Part 2)**
> > >
> > > **Summarize Task Properties**
> > >
> > > The properties of tasks suitable for action-repetition based framework have been actively researched in previous works from multiple perspectives. In the following, we try to summarize the properties of tasks that favor the proposed method. Note that only one property is needed in applications.
> > >
> > > **(a) Properties of tasks that favor Action Repetition (AR) Framework:**
> > >
> > > 1. **The decision frequency of MDP modeling is set larger than the minimum value to solve the task  [1,2].**
> > >    - **Analysis:** the action frequency of tasks can be difficult to be set exactly as the minimum value that doesn't comprise optimal control. Such frequency gap can be mined through AR framework automatically, leading to higher training efficiency with superior performance.
> > > 2. **The minimum optimal control frequency of the task varies during the episode [3,4].**
> > >    - **Analysis:** In MDP, the decision frequency is set *fixed* during interaction, while the minimum control frequency may *vary* across task phases. In simpler phases requiring lower frequencies , MDP frequency is set larger than needed, leaving room for action repetitions.
> > >    - **Examples:** robot locomotion on multiple terrains.
> > > 3. **Policies are preferred to take similar (possibly not identical) actions in most consecutive steps [5].**
> > >
> > > - **Analysis:**  AR encourages action persistence during *exploration*, which enables policies to explore the task more deeper and faster with fewer interactions, resulting in higher sample efficiency [2,6].
> > >
> > > - Note that this point is still effective even the optimal policy has no repeated actions. In such cases,  AR such as our method achieves higher sample efficiency through persistent exploration, and adjusts repetition rate automatically to achieve higher final reward.
> > >
> > > - **Examples:**  path planning, auto driving, robot manipulation, etc.
> > >
> > > **(b) Properties of tasks that favor Spatially Decoupled Action Repetition:**
> > >
> > > 1. **Tasks described in (a);**
> > >    - Reasons and examples are described in (a).
> > > 2. **Each action dimension requires different persistence with distinct control frequencies**
> > >    - **Analysis:** In MDP, the decision frequency is set *same* for each dimension. There may exist "redundant" frequency for some dimensions, leaving room for repetition at corresponding dimensions. Such repetition schema needs to be implemented in a spatially decoupled manner.
> > >    - **Examples:** various robotics control tasks, including  *Walker2d* shown in Fig. 4(b).
> > > 3. **The task focuses on different action dimensions at different task phases**
> > >    - **Analysis:** In some phases, some key dimensions are required to be decided more frequently than others. In other phases, the key dimensions may change, where different dimensions are decided more frequently. Such cases prefer to adjust repetition schemas for each dimension during the episode automatically.
> > >    - **Examples:** Humanoid robot / Mobile robot manipulation tasks, which focus on legs/chassis during moving, while focus on arms during manipulation.
> > >
> > > Fortunately, common continuous control tasks in practice are suitable for repetition-based DRL methods, thus has been actively researched/utilized in previous works [1-4]. In this work, experiment results on multiple scenarios also demonstrate the effectiveness empirically.
> > >
> > > Quantifying these properties is a challenging topic in this field. To our knowledge, it has not been fully solved in previous works on temporal abstraction/action repetition. We believe that there exist some potential metrics to quantify these properties. The following is one possible metric:
> > >
> > > $g = \max_{\{t,i\}}f_{t,i} - \frac{1}{T|A|} \sum_{t,i} f_{t,i}$ , where $f_{t,i}$ denotes the minimum optimal frequency of $i$-th dimension at time $t$, and $T$ denotes episode length.  $g$ is the difference between the highest frequency of each dimension during the episode and the average frequency, where bigger values indicate that the task is more suitable for the proposed method. This metric may be difficult to compute in some complex tasks, where $f_{t,i}$ may be estimated based on the policies.
> > >
> > > Thank you for your valuable feedback and insightful suggestions. In future, we will conduct further studies to clarify and quantify the task properties suitable for this method.
> > >
> > > [1] Grigsby et al., Towards automatic actor-critic solutions to continuous control, NeurIPS 2021
> > >
> > > [2] Yu, et al., Taac: Temporally abstract actor-critic for continuous control, NeurIPS 2021
> > >
> > > [3] Karimi et al. Dynamic Decision Frequency with Continuous Options. IROS 2023
> > >
> > > [4] Biedenkapp et al. TempoRL: Learning when to act. ICML 2021
> > >
> > > [5] Christmann et al. Benchmarking Smoothness and Reducing High-Frequency Oscillations in Continuous Control Policies. IROS 2024
> > >
> > > [6] Dabney et al., Temporally-Extended epsilon-Greedy Exploration, ICLR 2021
> > >
> > > [7] Lee et al., Reinforcement learning for control with multiple frequencies, NeurIPS 2020

---

> > > > ### Comment · Reviewer_sXke · 2024-12-02
> > > >
> > > > Thanks for the experiments. However, I would say the performance gain is marginal- probably 2 or 3 %.
> > > >
> > > > The follow up comment is more interesting.
> > > > > Examples: various robotics control tasks, including Walker2d shown in Fig. 4(b).
> > > >
> > > > I am still thinking why the proposed method works better in Walker2d. In general walking, it sounds odd that each joint moves at different frequencies- they are more or less synchronized with different phases.
> > > >
> > > > Anyway, I acknowledge the hardworking of the authors. I don't think I would change the score at this moment. However, I would keep all the feedback in my mind during the discussion phase. Thanks!

---

> > > > > ### Author Response · Authors · 2024-12-03
> > > > > **Response to Reviewer sXke**
> > > > >
> > > > > Thanks for your detailed review and valuable response.
> > > > >
> > > > > > However, I would say the performance gain is marginal- probably 2 or 3%.
> > > > >
> > > > > As you have pointed out, this task (tracking sinusoidal trajectory) is unsuitable for our method. However, SDAR still solves the task successfully with comparable performance with SAC. Besides, this env is simple for vanilla SAC, which solves this task with a few samples (about 30k) and **near-optimal score 195 (optimal score: 200)**, leaving limited space for performance improvement.
> > > > >
> > > > > > In general walking, it sounds odd that each joint moves at different frequencies.
> > > > >
> > > > > This moving pattern has been employed in some previous work on RL-based continuous control tasks. For example, Action Persistent Actor-Critic [1] sets different action frequencies for different joints manually in locomotion tasks, where action frequency for the thighs, the legs, and the feet are set to 1, 2, and 4 respectively.
> > > > >
> > > > > In this work, SDAR tries to find suitable persistence patterns for each task automatically. As described in Temporally Extended Epsilon Greedy [2], such policies with temporal persistence can improve ability to escape local optima, and achieve deeper explorations with less samples, leading to faster exploration and higher training efficiency.
> > > > >
> > > > > Thanks for your valuable response. If you have further questions or suggestions, please do not hesitate to let us know.
> > > > >
> > > > > [1] Lee, Jongmin, et al., Reinforcement learning for control with multiple frequencies, NeurIPS 2020
> > > > >
> > > > > [2] Dabney et al., Temporally-Extended epsilon-Greedy Exploration, ICLR 2021

---

### Official Review · Reviewer_NAN2 · 2024-11-03

**Soundness:** 4
**Presentation:** 4
**Contribution:** 3
**Rating:** 8
**Confidence:** 4

**Summary:**

A version of state-conditioned action-repeat is proposed that also allows
for arbitrary subsets of the action dimensions to be repeated, while the rest are
directly controlled by the policy. The paper describes in detail how to learn the
action-repeat-selection policy to be learned.  To cope with the combinatorial complexity
issue that arises for high-D action spaces, an importance-based sampling approach is further proposed.
The method is evaluated on a reasonably-sized set of continuous action problems,
ranging from low-D to high-D action spaces. The method is shown to be more sample efficient
than a reasonably-chosen set of model-free alternatives, including SAC and a variety
of other action-repeat-learning baselines.  Overall, the proposed method is shown to excel.
The learned patterns of action-subset-repeats are also visualized, to help provide
a qualitative understanding of their usage over time by the learned policies.

**Strengths:**

Strengths:
- clean idea, good execution, clearly written
- a "relatively simple" idea, entirely in the good sense, and so likely to have impact
- will inspire follow-up work;  learning structure in action representations remains
  very much an underexplored area, compared to learning structured or latent state representations

**Weaknesses:**

Weaknesses:
- I suspect that one could find an example where there is a price to be paid for using this approach,
  as opposed to it always being more sample efficient
- it is always hard to see if equal time has been spent tuning all the comparison RL algos;
  with the right setup and high-replay ratios, some of the RL algos can possibly see further improved sample efficiencies
- it's clear why we should care about learning rate / sample efficiency, but not so clear why
  we should actually care about action persistence of action fluctuation.

**Questions:**

Questions:
- how does experience replay sampling work when a- will be different?
  Or is a- explicitly stored with each tuple in the replay buffer?
- Could the learned "synergies" potentially be transferred when learning a new task?
- Perhaps I missed it, but are the exploration rates for the repetition schema policy, and the regular policy, being annealed?  Might that make sense?
- Ideally a method would also work when the "repetition" substructure could also be discovered even after the action-space was rotated, i.e., imagine all actions spanned all joints, to differing degrees, but the latent structure would still be there. Just a suggestion for something that would be fun (but arguably difficult) to pursue!

Minor feedback:
- At the end of the introduction, the second and third listed contributions probably belong together, as they
  describe the same thing. I like the visualization as another minor contribution.
- The notation might be easier to understand if the repetition schema is called a mask, m, rather than the current "b".
- Is it possible to collapse the two policies to a single deterministic policy pi, after training?
  This would be of important practical importance.
- "This demonstrates the necessity and effectiveness":  necessity is perhaps too strong;
  it is effective in terms of speeding learning, but arguably not "necessary"
- Table 2: "Avearage"  (typo)
- lines 479-480: "rotors" --> "joints"?
- line 483:  "Besides,"  suggest to remove

---

> ### Author Response · Authors · 2024-11-16
> **Response to Reviewer#NAN2 (Part 1)**
>
> Thank you for your uplifting review and valuable feedback. We appreciate your comments and would like to address them below.
>
> ## Weaknesses:
>
> > 1. I suspect that one could find an example where there is a price to be paid for using this approach, as opposed to it always being more sample efficient
>
> This is an interesting perspective and we agree with this at present. Each RL method is suitable for some tasks while maybe invalid in other scenarios. We will analyze this question using the following subquestions:
>
> 1. **Why SDAR achieves higher sample efficiency than vanilla DRL and previous repetition methods ?**
>
> - Previous RL studies have pointed out the significant influence of action persistence on training efficiency of RL methods [1-5]. It is important to take proper trade-off between action persistence and diversity during training[2].
> - Previous repetition works aim to improve action persistence in a flexible manner: Repeat for given steps (frame-skipping / N-rep in this work) -> Open-loop repetitions (TempoRL, UTE) -> Closed-loop methods (TAAC)
> - SDAR achieves a more flexible repetition strategy in spatially decoupled manner. It improves the balance between persistence and diversity.
>
> 2. **Why improving action persistence is effective for common task scenarios?**
>
> There exist multiple reasons for this phenomenon:
>
> - Improving action persistence leads to deeper exploration with faster training efficiency in common decision tasks [1-4]. Commonly practical tasks prefer policies to output similar actions in most successive steps, such as path planning.
> - In MDP, the decision frequency is fixed, which is difficult to be set exactly as the minimum value that doesn't comprise optimal control while leaving no room for temporal abstraction[4]. Besides, the optimal control frequency may vary during the whole episode, leaving room for action repetitions.
> - In tasks with |A|>1, each action dimension commonly requires different persistence with distinct control frequencies [5], such as robot joint controllers operating at different frequencies.
>
> 3. **What types of tasks are suitable or unsuitable for SDAR algorithm?**
>
> Based on Q2, we can find that repetition-based methods, such as our method, are suitable for common tasks in practical scenarios. E.g. path planning, robot locomotion, manipulation tasks, autonomous driving, etc.
>
> The following are potential tasks that are unsuitable for SDAR. **All requirements are needed to meet.**
>
> - The task prefers policies without any persistence at each action dimension during interaction. In other words, the policy needs to change motions dramatically for each dimension at each moment.
> - The decision frequency of MDP is set exactly as the minimum value to solve the task, leaving no room for temporal abstraction.
> - The minimum decision frequencies of each action dimension are same, and keep static during the episode.
>
> Above all, common tasks and MDP modeling in practical scenarios are suitable for repetition-based RL such as SDAR in this work.
>
> Thank you for pointing out this issue. In future works, we will conduct more research to identify the application scenarios suitable for our method.
>
> [1] Dabney, Will, et al., Temporally-Extended epsilon-Greedy Exploration, ICLR 2021
>
> [2] Yu, Haonan, et al., Taac: Temporally abstract actor-critic for continuous control, NeurIPS 2021.
>
> [3] Metelli, Alberto Maria, et al., Control Frequency Adaptation via Action Persistence in Batch Reinforcement Learning, ICML 2020.
>
> [4] Grigsby, Jake, et al., Towards automatic actor-critic solutions to continuous control, NeurIPS 2021.
>
> [5] Lee, Jongmin, et al., Reinforcement learning for control with multiple frequencies, NeurIPS 2020.

---

> ### Author Response · Authors · 2024-11-16
> **Response to Reviewer#NAN2 (Part 2)**
>
> > 2. It is always hard to see if equal time has been spent tuning all the comparison RL algos; with the right setup and high-replay ratios, some of the RL algos can possibly see further improved sample efficiencies
>
> In order to evaluate the performance of each method fairly, we try to avoid the influence of hyperparameter tuning.
>
> - SDAR shares same hyper-parameters with SAC in this work, including learning rate, hidden layer structures of networks, replay ratios, buffer sizes, soft update coefficient, etc. For other methods such as TAAC, we utilize parameters recommended by authors. Besides, we also tried to tune these parameters such as network sizes and soft update coefficients, which cannot achieve higher performance than official setups.
> - We agree that further tuning such as increasing replay ratios may achieve higher performance of each method. It is difficult to achieve essential improvement in performance. Thanks for your valuable feedback. We will try to improve performance of each method in future works with better setups and implementations.
>
>
>
> > 3. It's clear why we should care about learning rate / sample efficiency, but not so clear why we should actually care about action persistence of action fluctuation.
>
> In this work, we utilize APR (Action Persistence Rate) and AFR (Action Fluctuation Rate) to evaluate persistence and fluctuation of each method (c.f. Sec. 5.3), both of which are utilized to analyze the repetition schemas of each method.
>
> 1. **Action Persistence Rate**
>    - As described in (Line 055-057) and response to weakness 1, proper trade-off between action persistence and diversity is significant for repetition-based methods. Thus, action persistence is a metric to analyze the effectiveness of the repetition framework [1,2]. In this work,  SDAR improves persistence through a more flexible repetition schema.
>    - Thus, APR results of each method are given in Table 2 and Table 3. SDAR achieves higher APR than TAAC generally, demonstrating the higher persistence of our method.
>    - In addition, as shown in Table 3, SDAR achieves different persistence at different dimensions, demonstrating the effectiveness  of the spatially decoupled design.
>
> 1. **Action Fluctuation Rate**
>    - As shown in (Line 430), AFR evaluates the mean amplitude of fluctuations between successive steps, which can be seen as a special metric to evaluate action persistence.
>    - Besides, reducing action fluctuation of DRL is also meaningful in practice, which has been researched recently [6,7]. As shown in Table 2, SDAR achieves lower AFR with superior performance than other methods generally.
>
> [6] Song, Xujie, et al., LipsNet: a smooth and robust neural network with adaptive Lipschitz constant for high accuracy optimal control, ICML 2023.

---

> ### Author Response · Authors · 2024-11-16
> **Response to Reviewer#NAN2 (Part 3)**
>
> ## Questions：
>
> > 1. How does experience replay sampling work when $a^-$ will be different? Or is $a^-$ explicitly stored with each tuple in the replay buffer?
>
> The previous actions $a^-$ are explicitly stored in the replay buffer. As shown in (Line 10, Algorithm 1), the previous action $a_{t-1}$ is stored into the buffer $\mathcal{D}$ during the training process. The $a_{t-1}$ is the $a^-$ during training $\pi$ and $\beta$ described in Eq. (7)-(9), Sec. 4.3. According to your question, we have revised the manuscript accordingly to clarify this issue (Line 281-283). Thank you for pointing out this issue.
>
> > 2. Could the learned "synergies" potentially be transferred when learning a new task?
>
> Yes, this is a quite interesting and promising direction. SDAR can be potentially be integrated with transfer, multi-task, and even meta learning, because the "synergies" learned by $\beta$ may be shared across various tasks with same agent design.
>
> For example, given a synergy learned by a robot to walk on a plane, it can be easily transferred to new locomotion tasks such as climbing stairs.
>
> Furthermore, we conjecture that some universal "synergy" patterns can be found through training an agent in various tasks. For instance, given muscle-driven robots such as "human", muscles are controlled in hierarchical manners, where some muscles are always change actions together[8].
>
> Thank you for your valuable suggestion. This is one potential  and interesting topic for future studies.
>
>
>
> [8] Zuo, Chenhui, et al. Self Model for Embodied Intelligence: Modeling Full-Body Human Musculoskeletal System and Locomotion Control with Hierarchical Low-Dimensional Representation, ICRA 2024
>
>
>
> > 3. Perhaps I missed it, but are the exploration rates for the repetition schema policy, and the regular policy, being annealed? Might that make sense?
>
> In this work, we use entropy-based exploration for both two policies, with temperature parameters tuned automatically similar to SAC. The detailed formulations are illustrated in Eq. (7) and Eq. (10). Thus, the exploration rates for slection policy $\beta$ is tuned automatically. We find that such method leads to excellent repetition schemas.
>
> In the experiment, we have tried to decay the learning rates of $\beta$ during the training process, aiming to obtain a more stable $\beta$ with faster convergence. However, this attempt obtains obvious improvement, thus is not used in this work.
>
> In future works, the decoupling design may be incorporated into other DRL, such as TD3 and PPO, where other exploration strategies are needed and the annealed exploration will be used. Thank you for your suggestion.
>
>
>
> > 4. Ideally a method would also work when the "repetition" substructure could also be discovered even after the action-space was rotated, i.e., imagine all actions spanned all joints, to differing degrees, but the latent structure would still be there. Just a suggestion for something that would be fun (but arguably difficult) to pursue!
>
> This is a quite interesting and promising idea! There may exist consistency between action space structures and the repetition schemas of the policy. In order to find this potential property, we may need to use action representation learning during training [9]. Besides, such property can potentially contribute to reducing difficulty of transferring and improving generality of continuous DRL policies.
>
> [9] Chandak, Yash, et al. Learning action representations for reinforcement learning, ICML 2019.

---

> ### Author Response · Authors · 2024-11-16
> **Response to Reviewer#NAN2 (Part 4)**
>
> ## Minor Feedback
>
> > 1. At the end of the introduction, the second and third listed contributions probably belong together, as they describe the same thing. I like the visualization as another minor contribution.
>
> Thanks for your valuable suggestions. We will revise this point in the final version according to this suggestion.
>
> > 2. The notation might be easier to understand if the repetition schema is called a mask, m, rather than the current "b".
>
> Thank you for your valuable suggestion. We will revise this expression in the final version for the convenience of reviewers and chairs during the discussion period.
>
>
>
> > 3. Is it possible to collapse the two policies into a single deterministic policy pi, after training? This would be of important practical importance.
>
> Yes, two policies can be collapsed into a single deterministic policy $\pi_{all}$ in practical deployment. One possible schema is
>
> - The  $\beta$ selects act-or-repeat by comparing outputs with 0.5 rather than sampling on Bernoulli distributions.
> - The $\pi$ output the mean action value predicted by the network, rather than sampling on Gaussian distributions.
> - In practical implementation, $\pi_{all}$ is a single network composed of the $\beta$ and $\pi$ sub-networks.
>
> > 4. "This demonstrates the necessity and effectiveness": necessity is perhaps too strong; it is effective in terms of speeding learning, but arguably not "necessary"
>
> We agree that "necessity" is not a suitable expression. According to your suggestion, we have revised the expression accordingly. Thank you for pointing out this issue.
>
> > 5. Table 2: "Avearage" (typo)
>
>  We have corrected this typo in the manuscript. Thank you for pointing out this issue.
>
> > 6. lines 479-480: "rotors" --> "joints"?
>
> Yes, the "rotors" means the actuators of the corresponding joints. We have replaced this word with "joints", which is clearer without ambiguity.
>
> > 7. line 483: "Besides," suggest to remove
>
> We have removed this word to improve readability. Thank you for your suggestions.

---

> ### Comment · Reviewer_NAN2 · 2024-11-24
> **re: rebuttal response**
>
> Thank you for the detailed reply. I leave my score unchanged (8).
>
> This work makes sound progress on the fundamental (and under-appreciated) problem related to the temporal aspects of action-space design. The visualization of the learned results shows the (sometimes-surprising) effectiveness of dimension-specific action repeat. This paper will not be the final word in the area, but it's results will promote followup questions and work.

---

> > ### Author Response · Authors · 2024-11-25
> > **Thanks for your valuable response**
> >
> > Thank you very much for your thoughtful feedback! This is the first work to take consideration of difference between action dimensions into temporal abstraction, which contributes a novel viewpoint for future studies on temporal abstractions and action-space design. Your foresighted suggestions are valuable in helping us improve the paper and point out promising directions for future studies.
> >
> > We sincerely appreciate your recognition of our efforts to address the concerns. If you have any further questions or suggestions, please do not hesitate to let us know.

---

### Official Review · Reviewer_VdNA · 2024-11-05

**Soundness:** 3
**Presentation:** 3
**Contribution:** 2
**Rating:** 6
**Confidence:** 3

**Summary:**

This paper introduces Spatially Decoupled Action Repetition (SDAR), a framework that allows reinforcement learning (RL) agents to make separate act-or-repeat decisions for each action dimension. The method consists of a two-stage process: first, a selection policy determines if each dimension should repeat, followed by an action policy that generates new actions for those dimensions that chose not to repeat. Experiments across several continuous control tasks demonstrates the framework's performance and sample efficiency over baseline methods.

**Strengths:**

* The proposed method decouples the action dimensions and achieves more flexible and diverse repetition strategies, while maintaining high action persistence.
* The paper provides a thorough comparison with multiple baseline methods, including SAC, TAAC, and UTE. This context clarifies the advantages of SDAR over both open-loop and closed-loop action repetition approaches.

**Weaknesses:**

* The method might be limited when the action dimension is high, which might cause computational overheads.
* The novelty of this paper appears limited, as TAAC has employed a similar two-stage act-or-repeat mechanism; the primary contribution here seems to be the separation of action dimensions.

**Questions:**

* Can TAAC be regarded as an ablation of the proposed method by treating all action dimensions as a whole for repetition? How about its switch policy?
* Would the proposed method be able to be applied to on-policy methods like PPO, where better sample efficiency would be more valuable?

---

> ### Author Response · Authors · 2024-11-16
> **Response to Reviewer#VdNA (Part 1)**
>
> Thank you for your review and the explicitly listed concerns. We hope that our answers will help to resolve the concerns.
>
> ## Weakness
>
> > 1. The method might be limited when the action dimension is high, which might cause computational overheads.
>
> In this work, **high-dimensional action space (large $|A|$) mainly causes the following challenge:**
>
> The selection policy $\beta$ needs measure the score of $2^{|A|}$ repetition schemas at each step. This makes the inference and training of $\beta$ with Eq. (8) computationally expensive.
>
> **The following methods are utilized to solve the above challenges:**
>
> 1. As shown in Fig. 2,  $\beta$ network  is designed to output $|A|$ probabilities, indicating the probability of repetition at each dimension, rather than output $2^{|A|}$ probabilities directly.
> 2. As illustrated in Sec. 4.3,  $\beta$ network is trained with importance sampling described in Eq. (9). This method only needs to train $\beta$ based on sampling several $b\sim\beta$,  thus is more computationally efficient than Eq. (8) with large $|A|$.
> 3. As described in Sec. 4.1.2, the network in $\pi$ is designed with only $|S|+|A|$ inputs, where redundant information in $b$ and $a^-$ is removed to reduce network size based on operations in Eq. (2)-(4), leading to lower computational costs.
>
> Besides, the experiment on Humanoid (|A|=17) is conducted to measure the performance of our method:
>
> **(1) Performance**
>
> | Humanoid                      | SAC          | N-rep        | TempoRL      | UTE          | TAAC         | SDAR (Ours)      |
> | ----------------------------- | ------------ | ------------ | ------------ | ------------ | ------------ | ---------------- |
> | Episode Return                | 6184$\pm$717 | 5074$\pm$310 | 1022$\pm$397 | 2595$\pm$334 | 7308$\pm$244 | **7483$\pm$288** |
> | AUC (Normalized)              | 0.797        | 0.795        | 0.181        | 0.330        | 0.869        | **1.0**          |
> | Action Persistence Rate (APR) | 1.00         | 4.00         | 5.27         | 5.69         | 1.21         | 1.67             |
>
> The training curves are shown in Fig. 3.(d). As described in the table and figure, our method achieves higher performance  on sample efficiency and episode return. Besides, SDAR obtains higher APR than TAAC, demonstrating that SDAR can achieve more flexible repetition than TAAC while maintaining action diversity.
>
> **(2) Computational burden**
>
> Experiments are conducted to evaluate the time cost for training. The experiments are conducted on same GPU hardware and software environments. Same hyper-parameters are utilized such as learning rate. Both methods are trained to obtain same performance as SAC.
>
> | Environment         | SAC                 | SDAR                |
> | ------------------- | ------------------- | ------------------- |
> | Humanoid (\|A\|=17) | 595min (3.0M steps) | 365min (1.5M steps) |
> | Ant (\|A\|=8)       | 350min (2.3M steps) | 286min (1.3M steps) |
> | Walker2d (\|A\|=6)  | 282min (1.5M steps) | 167min (700K steps) |
>
> As shown in the table, SDAR requires both less time cost and step cost than SAC. More results on training time are given in (Table 6, Appendix B.4).
>
> Thank you for pointing out this issue. In future works, we aim to reduce computation and time costs of SDAR. For example, both $\beta$ and $\pi$ in SDAR can be optimized in parallel, which may achive higher time efficiency during training.

---

> ### Author Response · Authors · 2024-11-16
> **Response to Reviewer#VdNA (Part 2)**
>
> > 2. The novelty of this paper appears limited, as TAAC has employed a similar two-stage act-or-repeat mechanism; the primary contribution here seems to be the separation of action dimensions.
>
> Our work cannot be considered as the simple combination of TAAC with separation of action dimensions. We have taken tried this design in several environments, leading to unsatisfying results even failed training. The following are  experiment results of episode rewards.
>
> |             | SAC            | TAAC           | SDAR (Ours)        | TAAC+seperation$^{[*]}$ |
> | ----------- | -------------- | -------------- | ------------------ | ----------------------- |
> | Walker2d    | 5305$\pm$367   | 5660$\pm$394   | **6028$\pm$406**   | 413$\pm$22              |
> | HalfCheetah | 13122$\pm$2877 | 11148$\pm$3921 | **15131$\pm$1279** | -109$\pm$54             |
> | Pusher      | -22.5$\pm$1.40 | -30.5$\pm$1.85 | **-21.3$\pm$1.26** | -46.4$\pm$2.41          |
>
> As described in the table, "TAAC+seperation" cannot converge to satisfying results, demonstrating that the simple combination cannot work ideally, and the design of SDAR algorithm is necessary.  Actually, although both SDAR and TAAC use "two-stage policy", there exist essential differences between two methods.
>
> **New motivation:**
>
> - SDAR is the first work to take consideration of difference between action dimensions into temporal abstraction, via spatially decoupled action repetition as the methodology.  Compared to previous repetition methods, SDAR aims to achieve more flexible repetitions with an **improved balance between action *persistence* and *diversity***.
> - TAAC is the first work that incorporates *closed-loop* temporal abstraction into the actor-critic framework via the two-stage policy. Compared to *open-loop* repetitions, TAAC is more flexible with higher performance.
>
> Above all, both works present novel improvements for temporal abstraction in RL. However, each work contributes a unique viewpoint on this topic.
>
> **New method:**
>
> 1. **New "two-stage" policy:**
>    - TAAC firstly make new decision $\hat{a}$ using *action policy*, then switch between old action $a^-$ and new action $\hat{a}$ using Q values.
>    - SDAR firstly choose repetition schema $b$ using *selection policy* $\beta$, then make decisions based on $s$, $a^-$, and $b$. This method  avoids computing Q values for $2^{|A|}$ times.
> 2. **New two-stage "policy":** Both $\beta$ and $\pi$ are designed to adapt to decoupled action repetition. E.g. $\pi$ is designed to reduce redundant input dimensions to reduce computations (Sec. 4.1.2). Selection policy $\beta$ is a network rather than sampling based on Q values in TAAC (Sec. 4.3 in TAAC paper).
> 3. **New "training" method:** In SDAR,  $\beta$ policy network is trained with Eq. (8) for small |A| or Eq. (9) for large |A| with importance sampling. In  TAAC, the switch policy is updated via training Q functions.
>
> **Experiment results:**
>
> - Compared to TAAC, our method achieves higher sample efficiency than TAAC, higher rewards than TAAC, with higher persistence than TAAC. More details are given in Fig. 3, Table 1, and Table 2.
> - In addition, as the visualization of repetition schemas in Fig. 4 and Table 3, our method can adjust action persistence for each dimension automatically, demonstrating the effectiveness of the decoupled design and training framework.
>
> **Future impact:**
>
> Compared to previous works, SDAR contributes a novel viewpoint for future studies for temporal abstractions. For example:
>
> - Given an agent with various task scenarios, OR multiple agents with similar structures, corresponding policies may share similar repetition schemas. Such similarity can be potentially found from these viewpoints, leading to higher efficiency and generality of DRL.
> - There commonly exist hierarchical structures in action space for practical continuous control scenarios such as human [1]. In previous works, such structure needs human-craft case-by-case design generally, which may be found automatically with spatially decoupled algorithms in future.
>
> [*] In "TAAC+seperation", we use selection policy design same as TAAC, i.e. sampling based on the Q values of possible repetition schemas across all dimensions. The selection method in TAAC relies heavily on Q values for all repetition schemas, which may cause convergence failures in decoupled repetition frameworks.
>
> [1] Zuo, Chenhui, et al., Self Model for Embodied Intelligence: Modeling Full-Body Human Musculoskeletal System and Locomotion Control with Hierarchical Low-Dimensional Representation, ICRA 2024

---

> ### Author Response · Authors · 2024-11-16
> **Response to Reviewer#VdNA (Part 3)**
>
> ## Questions
>
> > 1. Can TAAC be regarded as an ablation of the proposed method by treating all action dimensions as a whole for repetition? How about its switch policy?
>
> No, TAAC is not an ablation of the new method by treating all action dimensions as a whole. As illustrated in response to weakness 2, there exist other differences between TAAC and SDAR, such as (1) the decision order of two-stage policies (2) the design of two policies (3) the training process of two policies. Thus, TAAC is not an ablation of the new method. In other words, SDAR is not same as TAAC + treating action dimensions separately.
>
>
> > 2. Would the proposed method be able to be applied to on-policy methods like PPO, where better sample efficiency would be more valuable?
>
> Yes, it is potential to be applied to on-policy methods such as PPO with some modifications. The following is one possible method:
>
> - The PPO policy is designed as two-stage policies depicted in Fig. 2.
>
> - The trajectory "buffer" $\mathcal{D}$ saves $(s,a,a^-,b,r,s')$, where $a^-$ denotes previous action, $b$ denotes the repetition schema during interactions.
>
> - Two policies are trained alternatively. Firstly, we can train $\pi$ using the objectives as follows:
>
> $$\max\_{\theta^\pi} \mathbb{E}\_{\tau\sim\mathcal{D}}\mathbb{E}\_{(s,a^-,b,a)\sim\tau}[\min(\frac{\pi(a|s,a^-,b)}{\pi\_{old}(a|s,a^-,b)}A(s,a),\;\; g(\frac{\pi(a|s,a^-,b)}{\pi_{old}(a|s,a^-,b)})A(s,a))],$$
>
> where $\tau$ denote trjectory, $A$ denotes advantage function, $g(x) = CLIP(x,1-\epsilon, 1+\epsilon)$ , and $\epsilon$ is a small hyperparameter. Similarly, the selection policy $\beta$ can be optimized as follows:
>
> $$
> \max\_{\theta^\beta} \mathbb{E}\_{\tau\sim\mathcal{D}}\mathbb{E}\_{(s,a^-,b,a)\sim\tau}[\min(\frac{\beta(b|s,a^-)}{\beta\_{old}(b|s,a^-)}A(s,a), g(\frac{\beta(b|s,a^-)}{\beta_{old}(b|s,a^-)})A(s,a))].
> $$
>
> Other processes keep same as the vanilla PPO algorithm. Besides, we need to notice that the main motivation of this work is achieving flexible repetitions with an improved balance between action *persistence* and *diversity*. This is the first work to take consideration of difference between action dimensions into temporal abstraction.
>
> Thank you for your valuable suggestion. This is one promising research direction for future works.

---

> ### Author Response · Authors · 2024-11-25
>
> Dear Reviewer VdNA,
>
> Thank you again for your time and efforts in reviewing this paper.
>
> Since the rebuttal period is ending soon, we would like to know if your concerns and questions have been addressed.  We would appreciate the opportunities to address any additional concerns you may have before the discussion phase ends.

---

> > ### Comment · Reviewer_VdNA · 2024-11-26
> >
> > Thank you for the detailed reply. I have increased my score to 6. Ablations of the proposed method SDAR like no action dimension separation would further make the results more convincing and interpretable.

---

> > > ### Author Response · Authors · 2024-11-28
> > >
> > > Thank you for your constructive feedback. According to your suggestion, we conduct ablation studies to evaluate the performance of SDAR without the decoupling design, requiring all dimensions to make the same choice during act-or-repeat selections.
> > >
> > > | Environment | Metric | SAC   | TAAC  | SDAR (Ours) | SDAR w/o decoupling |
> > > | ----------- | ------ | ----- | ----- | ----------- | ------------------- |
> > > | Humanoid    | Reward | 6184  | 7308  | **7483**    | 6219                |
> > > |             | AUC    | 0.797 | 0.869 | **1.0**     | 0.793               |
> > > | HalfCheetah | Reward | 13122 | 11148 | **15131**   | 13549               |
> > > |             | AUC    | 0.706 | 0.879 | **1.0**     | 0.930               |
> > > | Walker2d    | Reward | 5305  | 5660  | **6028**    | 5326                |
> > > |             | AUC    | 0.815 | 0.624 | **1.0**     | 0.870               |
> > > | Ant         | Reward | 5098  | 5509  | **6236**    | 5761                |
> > > |             | AUC    | 0.760 | 0.823 | **1.0**     | 0.829               |
> > > | Pusher      | Reward | -22.5 | -30.5 | **-21.3**   | -21.72              |
> > > |             | AUC    | 0.916 | 0.940 | **1.0**     | 0.980               |
> > >
> > > As described in the table, compared to the complete SDAR algorithm, SDAR w/o decoupling achieves inferior performance on both rewards and AUC, demonstrating the effectiveness of decoupling framework used in this work.
> > >
> > > Thanks for your detailed review and insightful suggestions. If you have any further questions or suggestions, please do not hesitate to let us know.

---

### Meta-Review · Area_Chair_T5Mo · 2024-12-21

**Metareview:**

This paper studies the problem of action repetition in RL and propose to decouple each dimension of the control when learning to act-or-repeat. Experiments in simulated continuous control tasks demonstrate the advantages of this approach over baselines.

Reviewer NAN2 commends the simplicity of the idea, noting that the paper “makes sound progress on the fundamental (and under-appreciated) problem related to the temporal aspects of action-space design.” Reviewer sXke raises questions regarding the assumptions underpinning the method and expresses concerns about the lack of clarity on the specific task properties that merit the performance. Despite reservations, reviewer sXke stated they would not oppose acceptance.

**Additional Comments On Reviewer Discussion:**

The authors clarified many questions during discussion and ran additional experiments to respond to the reviewers' concerns.

---

### Decision · Program_Chairs · 2025-01-22

Accept (Poster)